

# Extracting the symmetries of nonequilibrium quantum many-body systems

Aleksandr N. Mikheev[1,2,3°], Viktoria Noel[1,2°], Ido Siovitz[2], Helmut Strobel[2], Markus K. Oberthaler[2] and Jürgen Berges[1]

**1** Institut für Theoretische Physik, Ruprecht-Karls-Universität Heidelberg,
Philosophenweg 16, 69120 Heidelberg, Germany
**2** Kirchhoff-Institut für Physik, Ruprecht-Karls-Universität Heidelberg,
Im Neuenheimer Feld 227, 69120 Heidelberg, Germany
**3** Institut für Physik, Johannes Gutenberg-Universität Mainz,
Staudingerweg 7, 55128 Mainz, Germany

## Abstract

Symmetries play a pivotal role in our understanding of the properties of quantum many-body systems. While there are theorems and a well-established toolbox for systems in thermal equilibrium, much less is known about the role of symmetries and their connection to dynamics out of equilibrium. This arises due to the direct link between a system's thermal state and its Hamiltonian, which is generally not the case for nonequilibrium dynamics. Here we present a pathway to identify the effective symmetries and to extract them from data in nonequilibrium quantum many-body systems. Our approach is based on exact relations between correlation functions involving different numbers of spatial points, which can be viewed as nonequilibrium versions of (equal-time) Ward identities encoding the symmetries of the system. We derive symmetry witnesses, which are particularly suitable for the analysis of measured or simulated data at different snapshots in time. To demonstrate the potential of the approach, we apply our method to numerical and experimental data for a spinor Bose gas. We investigate the important question of a dynamical restoration of an explicitly broken symmetry of the Hamiltonian by the initial state. Remarkably, it is found that effective symmetry restoration can occur long before the system equilibrates. We also use the approach to define and identify spontaneous symmetry breaking far from equilibrium, which is of great relevance for applications to nonequilibrium phase transitions. Our work opens new avenues for the classification and analysis of quantum as well as classical many-body dynamics in a large variety of systems, ranging from ultracold quantum gases to cosmology.

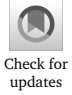

## Contents

---

° These authors contributed equally to the development of this work.

# 1   Introduction and overview

Important progress in our understanding of the complexity of macrophysics in quantum many-body systems has been achieved with classical computers for ground state or equilibrium properties. However, ab initio understanding of general dynamical or nonequilibrium behavior is particularly scarce in situations that are not simple extensions of near-equilibrium properties, such as the emergence of instabilities or turbulent flows. The search for emergent theories that effectively describe nonequilibrium macroscopic behavior, their classification and justification from first principles is one of the most pressing research directions in quantum many-body physics [1–3].

The notion of effective field theories for macroscopic behavior is a well-established powerful tool for equilibrium many-body systems, where the symmetries of the underlying Hamiltonian or action together with the knowledge of the order-parameter field allows one to construct the relevant description consistent with the symmetries [4,5]. However, out of equilibrium it is especially important to distinguish the symmetries of a state from the symmetries of a Hamiltonian. In fact, even the simplest nonequilibrium states with an order-parameter field that is initially not in its free-energy ground state can explicitly break a symmetry of the Hamiltonian in general. This raises the important question about the effective or emergent symmetries of nonequilibrium systems and whether/when explicitly broken symmetries get dynamically restored. To address this question, one needs to be able to quantify the symmetry content of a nonequilibrium state. This is also a crucial prerequisite for extracting the effective field theory actions [6,7] or Hamiltonians [8] from experimental or simulation data of quantum many-body systems. The question of dynamical symmetry restoration has been recently investigated based on entanglement asymmetry [9,10] and single-body density matrix [11].

In this work we describe a general pathway for extracting the effective symmetries of nonequilibrium quantum many-body systems using equal-time correlation functions. The approach takes into account that the density operator $\hat{\rho}_t$ describing a nonequilibrium state at any time $t$ may not be directly related to the Hamiltonian $\hat{H}$, unlike in thermal equilibrium, where $\hat{\rho}_{\text{eq}} \sim \exp(-\beta \hat{H})$ for the example of a canonical ensemble. Instead, we exploit that the symmetries can be classified on the level of observables, i.e., expectation values $\text{Tr}[\hat{\rho}_t \, \hat{\mathcal{O}}(\boldsymbol{x}_1, \ldots, \boldsymbol{x}_n)]$

of $n$-point operators $\hat{\mathcal{O}}(\boldsymbol{x}_1, \ldots, \boldsymbol{x}_n)$. We derive exact relations between expectations values of operators involving different numbers $n$ of spatial points, which encode the symmetry properties of the system. Our equations can be viewed as nonequilibrium versions of (equal-time) Ward identities [12]. For the example of a spin-one Bose gas, we show that extracting the $n$-point functions from spatially resolved data allows one to efficiently uncover the presence or absence of a given symmetry. For this, we define symmetry witnesses and apply our approach to analyze the dynamical effective restoration of explicit symmetry breaking. Remarkably, we observe that effective symmetry restoration can occur long before the system equilibrates, which is a crucial ingredient for the construction of effective theories for nonequilibrium evolutions. Importantly, we also demonstrate how the method can be used to define and identify spontaneous symmetry breaking even far from equilibrium, opening up numerous applications for nonequilibrium phase transitions.

While the approach can be used for any analytical or classical simulation technique of quantum many-body systems, we emphasize that it is particularly well suited for large-scale (analog) quantum simulations based on setups with ultracold quantum gases [13,14]. These systems can realize a wide range of Hamiltonians with different symmetries, variable interactions and degrees of freedom based on atomic, molecular, and optical physics engineering. They offer high control in the preparation and read-out of the quantum dynamics, with the ability to explore new regimes even far from equilibrium [15–17] that are otherwise difficult to access directly.

The paper is organized as follows. We start in Sec. 2 with a general discussion on symmetries in many-body systems, highlighting the differences between equilibrium and nonequilibrium cases. Taking an ultracold spinor Bose gas in one spatial dimension as an example, which is described in Sec. 3, we demonstrate our method in Sec. 4 and derive symmetry identities between equal-time $n$-point functions and symmetry witnesses. We then prepare such a system in a state with explicit symmetry breaking and investigate the subsequent evolution using classical-statistical simulation methods in Sec. 5. In Sec. 6, we consider experimental data for the spinor Bose gas and apply it to the analysis of spontaneous symmetry breaking out of equilibrium. We end with a conclusion and outlook in Sec. 7.

## 2 Symmetries and dynamics

For the following, it will be important to distinguish symmetries of a state or density operator from symmetries of the Hamiltonian that governs the equations of motion [18]. A Hamiltonian $\hat{H}$ is symmetric under the group of transformations $G$ if $[U, \hat{H}] = 0$ for every $U \in G$. This group can be either discrete or continuous, with $U$ forming an (anti-)unitary representation of $G$ on the Hilbert space of the system [19,20]. In this work, we focus on the case of continuous unitary symmetries. In addition, we assume that the considered continuous symmetries have the structure of a Lie group, whose elements can be written as

$$U = \exp(\mathrm{i}\alpha_k Q_k), \qquad [Q_i, Q_j] = \mathrm{i}f_{ijk}Q_k, \tag{1}$$

where $f_{ijk}$ are the structure constants that characterize the underlying Lie algebra, and the operators $Q_k$ are the generators of the group. For brevity we have restricted ourselves to elements of $G$ that are simply connected to the unity element. Since $U$ is unitary, the operators $Q_k$ are Hermitian and taken to correspond to physical observables. From Eq. (1) it immediately follows that $[Q_k, \hat{H}] = 0$, implying that the generators of $G$ are conserved quantities.

On the other hand, the state at time $t$ described by the density operator $\hat{\rho}_t$ is symmetric under $G$ if $[U, \hat{\rho}_t] = 0$ for every $U \in G$. From this, one also concludes the following rigorous property for the unitary time evolution of quantum systems described by the von Neumann

equation: if the density operator $\hat{\rho}_{t_0}$ explicitly breaks a symmetry of the Hamiltonian $\hat{H}$ at some given time $t_0$, then it cannot be restored on a fundamental level at any other time. Conversely, starting with a symmetric state and following a unitary evolution respecting the same symmetry, it will never be explicitly broken.

However, these strict statements are not in conflict with the assertion that typical observables may show emergent phenomena which involve the effective restoration of an initially broken symmetry or vice versa. In this work, we will consider expectation values $\mathrm{Tr}\big[\hat{\rho}_t \hat{\mathcal{O}}(\boldsymbol{x}_1,\ldots,\boldsymbol{x}_n)\big]$ of $n$-point operators $\hat{\mathcal{O}}(\boldsymbol{x}_1,\ldots,\boldsymbol{x}_n)$ as observables. An effective symmetry still remains a set of transformations which leave observable properties of the system unchanged, though the set of observables becomes restricted in practice, which in our case will be related to finite numbers for $n$. For instance, the notion of effective or relevant symmetries for observable properties is at the heart of macroscopic theories for nonequilibrium evolutions, such as effective kinetic theories or hydrodynamics describing the long-time and long-distance behavior of an underlying microscopic many-body system in terms of few-point functions only [2]. In this respect, the discussion also closely resembles the one concerning thermalization in closed quantum systems with unitary time evolution [1].

So far we have distinguished the symmetries of the state from the those of the Hamiltonian with the possibility of explicit symmetry breaking. However, for many-body systems it is also important to distinguish an explicit breaking of a symmetry from the phenomenon of spontaneous symmetry breaking. The latter is crucial, e.g., for our understanding of typical phase transitions where an order-parameter can be defined to vanish on one side of the transition while taking on a nonzero value otherwise. Though this is of course well established in equilibrium, the definition and detection of spontaneoulsy broken symmetries out of equilibrium is much less explored.

Spontaneous symmetry breaking implies that the symmetry of the system's state is reduced to a residual symmetry subgroup of $G$ without explicit symmetry violation. Generally, the system will be in a superposition of degenerate states such that the symmetry breaking is not manifest. To efficiently characterize spontaneous symmetry breaking in terms of an order parameter, one needs to lift the degeneracy and favor one of the infinitely many symmetry-breaking configurations. This is typically achieved by adding a small symmetry-breaking perturbation to the Hamiltonian, such as $\hat{H} \rightarrow \hat{H} + \int J\,\hat{\mathcal{O}}$ for a given order-parameter operator $\hat{\mathcal{O}}$. To remove the explicit symmetry breaking in the end, such a bias is introduced as a limiting procedure. Spontaneous symmetry breaking is then identified by a nonvanishing expectation value

$$\lim_{J\to 0^+} \mathrm{Tr}\big[\hat{\rho}_t\,\hat{\mathcal{O}}(\boldsymbol{x})\big] = v_t(\boldsymbol{x})\,. \tag{2}$$

Crucially, in the case of spontaneous symmetry breaking one finds a nonzero order parameter, $v_t(\boldsymbol{x}) \neq 0$, even in the limit of a vanishing perturbation, $J \to 0^+$. On the other hand, $v_t(\boldsymbol{x})$ is zero in the symmetric state. The choice of an order parameter operator is not unique, although often suggested by the physics of the spontaneous symmetry breaking. Here, we have restricted ourselves to cases that can be characterized by a local order parameter. For translationally invariant systems in space and/or time, the function $v_t(\boldsymbol{x})$ naturally reduces to a respective constant.

For nonequilibrium systems, there are interesting further options to introduce a symmetry-breaking bias, e.g., through the choice of an explicit symmetry-breaking state at a given initial time $t_0$ with

$$[U,\hat{\rho}_{t_0}] \neq 0\,, \qquad [V,\hat{\rho}_t] = 0\,, \tag{3}$$

while the symmetry of the Hamiltonian remains unaffected with $[U,\hat{H}] = 0$. In this case, the initial explicit symmetry breaking is not restricted to small perturbations. In situations where the explicitly broken symmetry gets effectively restored dynamically during the time

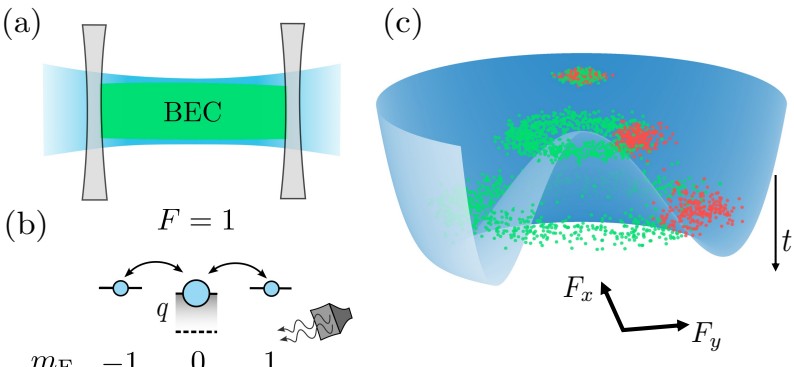

Figure 1: (a) Quasi one-dimensional BEC in a box-like potential formed by an elongated dipole trap (blue) with repulsive walls (light grey). The overall density (green) is approximately uniform throughout the cloud. (b) The effective quadratic Zeeman energy difference $q$ between the $m_F = 0$ and $m_F = \pm 1$ levels is adjusted using off-resonant microwave dressing (grey). This enables the tuning of spin-changing collisions into resonance, which can redistribute population among the hyperfine levels. (c) Schematics of many individual realizations averaged over in green, with a single realization highlighted in red in a "sombrero" potential associated with spontaneous symmetry breaking. The vertical $t$ arrow indicates the evolution in time.

evolution, spontaneous symmetry breaking is still signalled by the emergence of a nonzero order parameter (2). Typically, this requires an evolution of the system to sufficiently late times such that the initial explicit symmetry breaking is effectively reduced to a small perturbation. In the following sections, we will employ and discuss how symmetry can be broken through initial conditions in systems out of equilibrium. Specifically, we will introduce relationships between different $n$-point functions to identify symmetries and to distinguish between explicit and spontaneous symmetry breaking.

## 3 Spinor Bose gas

Both experimentally and in our numerical simulations, we consider a homogeneous one-dimensional spin-1 Bose gas described by the Hamiltonian [21]

$$\hat{H} = \int dx \left[ \hat{\psi}_m^\dagger \left( -\frac{1}{2M} \frac{\partial^2}{\partial x^2} + q f_z^2 \right) \hat{\psi}_m + \frac{c_0}{2} :\hat{n}^2: + \frac{c_1}{2} :\hat{\boldsymbol{F}}^2: \right], \tag{4}$$

where $\hat{\boldsymbol{\psi}} = (\hat{\psi}_1, \hat{\psi}_0, \hat{\psi}_{-1})^T$ is the three-component bosonic field representing the magnetic sub-levels $m_F = 0, \pm 1$ of the $F = 1$ hyperfine manifold, $M$ denotes the atom mass, and $\hat{n} = \hat{\psi}_m^\dagger \hat{\psi}_m$. The spin-changing collisions are described in terms of the spin operators $\hat{F}_i = \hat{\psi}_m^\dagger (f_i)_{mm'} \hat{\psi}_{m'}$, with $\boldsymbol{f} = (f_x, f_y, f_z)^T$ being the generators of the so(3) Lie algebra in the three-dimensional fundamental representation. The bosonic field operators obey the standard commutation relations $[\hat{\psi}_m(x), \hat{\psi}_{m'}^\dagger(x')] = \delta_{mm'}\delta(x - x')$, $[\hat{\psi}_m(x), \hat{\psi}_{m'}(x')] = 0$. Together with $[f_i, f_j] = i\varepsilon_{ijk} f_k$, this readily implies $[\hat{F}_i(x), \hat{F}_j(x')] = i\varepsilon_{ijk}\hat{F}_k(x)\delta(x - x')$. Here and in the following, Einstein's summation convention is implied and we use units where $\hbar = k_B = 1$.

Experimentally, we will apply our analysis to measurements from a spinor Bose–Einstein condensate of $^{87}$Rb, which features rotationally invariant ferromagnetic ($c_1 < 0$) spin-spin as well as repulsive ($c_0 > 0$) density-density interactions, with $|c_0/c_1| \approx 200$. This condensate is

confined in a quasi-one-dimensional box trap, as illustrated in Fig. 1(a). The quadratic Zeeman shift $q$ is induced by an external magnetic field which shifts the energy of the $m_{\mathrm{F}} = \pm 1$ levels relative to the $m_{\mathrm{F}} = 0$ component, and is adjusted by using off-resonant microwave dressing, as depicted in Fig. 1(b). We will consider data where the system is initialized with zero average longitudinal ($z$-axis) spin such that only the $m_F = 0$ sublevel is populated. The microwave dressing initiates the spin-exchange dynamics, and excitations build up in the $F_x - F_y$ plane, with the spin acquiring a mean length with a random orientation in the $F_x - F_y$ plane. This transversal spin degree of freedom is examined by the spatially resolved detection of the complex valued field $F_\perp(x) = F_x(x) + iF_y(x)$. Experimentally, we simultaneously extract the spatial spin profiles $F_x(x)$ and $F_y(x)$ via spin rotations from the $F_x - F_y$ plane to the $F_z$ direction and subsequent absorption imaging [16, 22]. For more details on the experimental setup and specific parameters, see App. A and Ref. [23].

The Hamiltonian (4) is symmetric under SO(2) × U(1) transformations for $q \neq 0$, where SO(2) denotes the group of rotations about the $F_z$ axis on the $F = 1$ hyperfine manifold. The spin operator $\hat{F}$ can play the role of the order parameter for the symmetry breaking of the SO(2) group. In the case of spontaneous symmetry breaking, according to the definition (2), there is a nonzero expectation value $\langle \hat{F}_i \rangle$. This situation is illustrated in Fig. 1(c). Due to the underlying SO(2) symmetry, we can always align the expectation value along one of the axes, e.g., $\langle \hat{F}_x \rangle = 0$, $\langle \hat{F}_y \rangle = v_t$.

Establishing long-range coherence across the entire system requires some time. This is especially true for lower-dimensional systems with continuous symmetries, where fluctuations preventing the build-up of long-range order are very strong, as highlighted by the Mermin–Wagner theorem [24]. To ensure an adequate level of coherence across the system during the time of observation, we reduce our analysis to a finite central region of our data as specified in App. A. For this subsystem, the condensate builds up a constant phase across the sample and the order parameter can be assumed to be approximately homogeneous for the considered evolution times.

# 4 Symmetry identities between equal-time correlation functions

We are probing the symmetry content of our system via equal-time correlation functions. Since such correlators can be extracted from measurements at different snapshots in time, they are particularly convenient for studying cold atom systems and matching theory to experiment. In spinor Bose gases, a convenient choice of experimentally accessible observables are spin operators $\hat{F}_i$. On a theoretical level, the corresponding equal-time correlation functions can then be conveniently extracted from the generating functional

$$Z_t[\boldsymbol{J}] = \mathrm{Tr}\left\{ \hat{\rho}_t \exp\left[ \int \mathrm{d}x\, \boldsymbol{J}(x) \cdot \hat{\boldsymbol{F}}(x) \right] \right\}, \tag{5}$$

where $\hat{\rho}_t$ is the density matrix of the system in the Schrödinger picture at time $t$, not necessarily normalized to unity. Symmetrically ordered equal-time spin correlation functions are obtained by taking derivatives with respect to $J_i(x)$ and setting the latter to zero:

$$\frac{Z^{(n)}_{t,i_1 \ldots i_n}[0](x_1, \ldots, x_n)}{Z_t[0]} = \frac{1}{n!} \sum_{\sigma \in S_n} \left\langle \hat{F}_{i_{\sigma_1}}(x_{\sigma_1}) \ldots \hat{F}_{i_{\sigma_n}}(x_{\sigma_n}) \right\rangle. \tag{6}$$

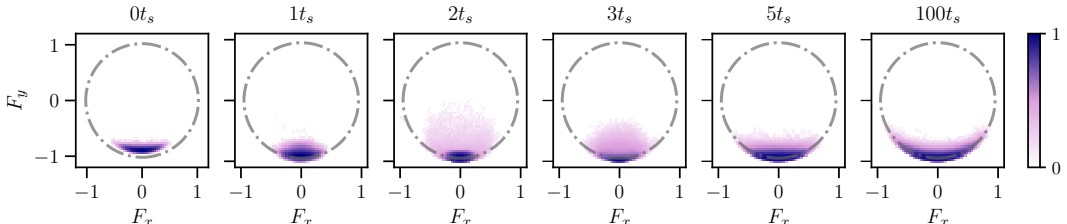

Figure 2: Histograms of the spin orientations in the $F_x - F_y$ plane normalized by the atom number for $q_f = 0.6n|c_1|$ and averaged over $10^3$ runs. The dash-dotted line represents the average spin length $\langle|F_\perp|\rangle = \sqrt{1 - (q/2n|c_1|)^2} \sim 0.95$.

Here, the prefactor $1/Z_t[0]$ takes care of the density matrix normalization, $S_n$ denotes the set of all permutations of $\{1, \ldots, n\}$, $\langle\ldots\rangle \equiv \mathrm{Tr}\{\hat{\rho}_t \ldots\}$, and we have introduced the notation

$$Z^{(n)}_{t, i_1 \ldots i_n}[\boldsymbol{J}](x_1, \ldots, x_n) \equiv \frac{\delta^n Z_t[\boldsymbol{J}]}{\delta J_{i_1}(x_1) \ldots \delta J_{i_n}(x_n)}. \tag{7}$$

The correlation functions (6) contain disconnected, lower-order parts. To remove this redundant information and generate connected correlation functions, one can invoke an equal-time equivalent of the Schwinger functional,

$$E_t[\boldsymbol{J}] = \log Z_t[\boldsymbol{J}]. \tag{8}$$

As an example, a two-point connected symmetric spin correlation function generated by the functional $E_t$ is given by

$$E^{(2)}_{t,xy}[0](x_1, x_2) = \frac{1}{2}\left\langle \hat{F}_x(x_1)\hat{F}_y(x_2) + \hat{F}_y(x_2)\hat{F}_x(x_1)\right\rangle - \left\langle \hat{F}_x(x_1)\right\rangle\left\langle \hat{F}_y(x_2)\right\rangle, \tag{9}$$

and correspondingly for higher-order correlation functions.

Since the spin operators $\hat{F}_i$ transform trivially under U(1), we will focus on the SO(2) part and derive associated symmetry identities between different correlation functions. Following the discussion in the previous sections, we will assume that the initial state $\hat{\rho}_{t_0}$ is also SO(2)-invariant, ensuring that the symmetry is fully respected on the dynamical level. In this case, the density matrix $\hat{\rho}_t$ remains formally symmetric at any time $t \geq t_0$, even in the case of spontaneous symmetry breaking. As pointed out above, to address the latter scenario, one has to introduce a symmetry-breaking bias to the system. In this work, the role of such a bias will be played by the sources $J_i$ coupled to the spin operators in the definition (5) of the generating functional, which will be addressed in more detail in the following.

From (5) we conclude together with $\hat{\rho}_t = U\hat{\rho}_t U^{-1}$, with $U \in$ SO(2), that

$$Z_t[\boldsymbol{J}] = \mathrm{Tr}\left\{\hat{\rho}_t \exp\left[\int \mathrm{d}x\, \boldsymbol{J}(x) \cdot \left(U^{-1}\hat{\boldsymbol{F}}(x)U\right)\right]\right\}, \tag{10}$$

where we have used the cyclic property of trace and $U^{-1}\exp(A)U = \exp(U^{-1}AU)$.

The spin operators $\hat{F}_i$ live in the fundamental representation of the rotation group and thus transform as

$$\hat{F}_i \to R_{ij}(\epsilon)\hat{F}_j = \hat{F}_i + i\epsilon T_{ij}\hat{F}_j + \mathcal{O}(\epsilon^2), \qquad T = \begin{pmatrix} 0 & i & 0 \\ -i & 0 & 0 \\ 0 & 0 & 0 \end{pmatrix}, \tag{11}$$

where $R(\epsilon)$ denotes the rotation matrix by an angle $\epsilon$ about the $F_z$ axis with its single generator $T$.

Together, Eqs. (5) – (11) imply $Z_t[\boldsymbol{J}] = Z_t[R^{-1}\boldsymbol{J}]$, and likewise $E_t[\boldsymbol{J}] = E_t[R^{-1}\boldsymbol{J}]$, where we have used the fact that $\boldsymbol{J} \cdot \left(R\hat{\boldsymbol{F}}\right) = \left(R^{-1}\boldsymbol{J}\right) \cdot \hat{\boldsymbol{F}}$. Taking $R$ to be infinitesimal, this yields $E_t[J_x - \epsilon J_y, J_y + \epsilon J_x] - E_t[J_x, J_y] = 0$. Expanding it to linear order in the rotation angle $\epsilon$, we finally derive the master symmetry identity:

$$\int \mathrm{d}x' \left[ J_x(x') E_{t,y}^{(1)}[\boldsymbol{J}](x') - J_y(x') E_{t,x}^{(1)}[\boldsymbol{J}](x') \right] = 0 . \tag{12}$$

By taking further $J$-derivatives one can generate an infinite hierarchy of symmetry identities encoding the SO(2) symmetry of the system.

Here and in the following, we assume that the mean field does not break spatial homogeneity. To emphasize the distinction between the fields $\hat{F}_x$ and $\hat{F}_y$, we then introduce the notation $(F_x, F_y) \to (\pi, \sigma)$, $(J_x, J_y) \to (J_\pi, J_\sigma)$, and accordingly $\langle \hat{\pi} \rangle = 0$ and $\langle \hat{\sigma} \rangle = v_t$. To allow for a spontaneous symmetry breaking scenario, we first explicitly break the symmetry via a linear source term $\int \mathrm{d}x\, J\, \hat{\sigma}(x)$, cf. the discussion in Sec. 2:

$$\langle \hat{\sigma} \rangle = \lim_{J \to 0^+} E_{t,\sigma}^{(1)}[J_\pi = 0, J_\sigma = J] = v_t . \tag{13}$$

The symmetry-breaking case corresponds to $v_t \neq 0$, whereas $v_t = 0$ in the symmetric phase. For spin systems, this symmetry-breaking term allows for a simple physical interpretation as a deformation of the initial density matrix, which is discussed in more detail in App. B.

Differentiating the master symmetry identity (12) once with respect to $J_\pi$ we get

$$\int \mathrm{d}x' \left[ \delta(x' - x'') E_{t,\sigma}^{(1)}[\boldsymbol{J}](x') + J_\pi(x') E_{t,\sigma\pi}^{(2)}[\boldsymbol{J}](x', x'') - J_\sigma(x') E_{t,\pi\pi}^{(2)}[\boldsymbol{J}](x', x'') \right] = 0 . \tag{14}$$

Setting the sources to $(0, J)$ and going to Fourier space we obtain

$$E_{t,\sigma}^{(1)}[0, J] - J\, \tilde{E}_{t,\pi\pi}^{(2)}[0, J](p = 0, -p = 0) = 0 , \tag{15}$$

where we have introduced the notation

$$E_{t,i_1 \dots i_n}^{(n)}(p_1, \dots, p_n) \equiv 2\pi \delta \left( \sum_{i=1}^{n} p_n \right) \tilde{E}_{t,i_1 \dots i_n}^{(n)}(p_1, \dots, p_n) . \tag{16}$$

Similarly, differentiating the master symmetry identity (12) once with respect to both $J_\pi$ and $J_\sigma$ and then setting the sources to $(0, J)$ yields

$$J \lim_{q \to 0} \tilde{E}_{t,\pi\pi\sigma}^{(3)}[0, J](q, p, -p - q) = \tilde{E}_{t,\sigma\sigma}^{(2)}[0, J](p, -p) - \tilde{E}_{t,\pi\pi}^{(2)}[0, J](-p, p) . \tag{17}$$

Taking the $J \to 0^+$ limit and using (13) and (15) we then find

$$v_t \lim_{q \to 0} \frac{\tilde{E}_{t,\pi\pi\sigma}^{(3)}(q, p, -p - q)}{\tilde{E}_{t,\pi\pi}^{(2)}(q, -q)} = \tilde{E}_{t,\sigma\sigma}^{(2)}(p, -p) - \tilde{E}_{t,\pi\pi}^{(2)}(-p, p) , \tag{18}$$

with $\tilde{E}_t^{(n)} \equiv \tilde{E}_t^{(n)}[J_\pi = 0, J_\sigma = 0]$. Here, we have taken into account that only the quotient of $\tilde{E}_{t,\pi\pi\sigma}^{(3)}(q, p, -p - q)$ and $\tilde{E}_{t,\pi\pi}^{(2)}(q, -q)$ may have a finite $q \to 0$ limit. While Eq. (18) connects

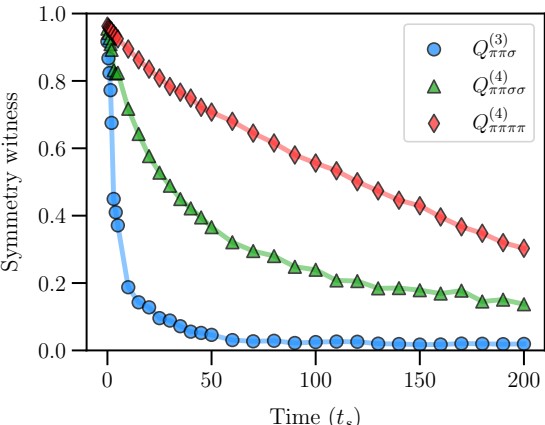

Figure 3: Evolution of the symmetry witnesses $Q^{(n)}$ for a system prepared in a state which explicitly breaks the SO(2) symmetry of the Hamiltonian with a subsequent quench from $q_i = 0.9n|c_1|$ to $q_f = 0.6n|c_1|$, where $0 \leq Q^{(n)} \leq 1$. The value of $Q^{(n)} = 0$ corresponds to the absence of explicit symmetry violation. Here, $Q^{(3)}_{\pi\pi\sigma}$ is the identity connecting two- and three-point functions appearing in Eq. (18), while $Q^{(4)}_{\pi\pi\sigma\sigma}$ and $Q^{(4)}_{\pi\pi\pi\pi}$ connect three- and four-point functions.

two- and three-point functions, additional symmetry identities relating higher-order correlation functions can be obtained by taking further derivatives:

$$v_t \lim_{k\to 0} \frac{\tilde{E}^{(4)}_{t,\pi\pi\sigma\sigma}(k,p,q,-k-p-q)}{\tilde{E}^{(2)}_{t,\pi\pi}(k)} = \tilde{E}^{(3)}_{t,\sigma\sigma\sigma}(p,q,-p-q) - \tilde{E}^{(3)}_{t,\pi\pi\sigma}(q,p,-p-q) - \tilde{E}^{(3)}_{t,\pi\pi\sigma}(p,-p-q,q),$$
(19a)

$$v_t \lim_{k\to 0} \frac{\tilde{E}^{(4)}_{t,\pi\pi\pi\pi}(k,p,q,-k-p-q)}{\tilde{E}^{(2)}_{t,\pi\pi}(k)} = \tilde{E}^{(3)}_{t,\pi\pi\sigma}(p,q,-p-q) + \tilde{E}^{(3)}_{t,\pi\pi\sigma}(p,-p-q,q) + \tilde{E}^{(3)}_{t,\pi\pi\sigma}(q,-p-q,p),$$
(19b)

and so forth.

Symmetry identities, akin to those derived in the present section, then serve as a manifestation of the system's symmetry properties on the level of correlation functions. Since $n$-point correlation functions can be readily extracted from numerically simulated data or experimental measurements, the symmetry identities can be explicitly checked. This makes them a powerful tool for analyzing the symmetry content of quantum many-body systems, allowing to determine whether the symmetry is broken explicitly, spontaneously, or not broken at all.

Based on the above symmetry identities one can introduce symmetry witnesses, which provide efficient measures of the symmetry content of a given system. In particular, higher-order correlation functions are often difficult to visualize and the introduction of a norm as a measure can be very convenient. Defining the left- and right-hand sides of (18) as

$$f^{(3)}_{t,\pi\pi\sigma}(p) = v_t \lim_{q\to 0} \frac{\tilde{E}^{(3)}_{t,\pi\pi\sigma}(q,p,-p-q)}{\tilde{E}^{(2)}_{t,\pi\pi}(q,-q)},$$

$$f^{(2)}_{t,\pi\pi\sigma}(p) = \tilde{E}^{(2)}_{t,\sigma\sigma}(p,-p) - \tilde{E}^{(2)}_{t,\pi\pi}(-p,p),$$
(20)

we may encode the symmetry content by measuring a distance between the two functions using the standard $L_1$-norm, $\|f\| = L^n \int dp_1 \ldots dp_n |f(p_1,\ldots,p_n)|$, with $L$ being the system size setting the smallest unit of momentum $1/L$. To avoid biasing the IR momentum region,

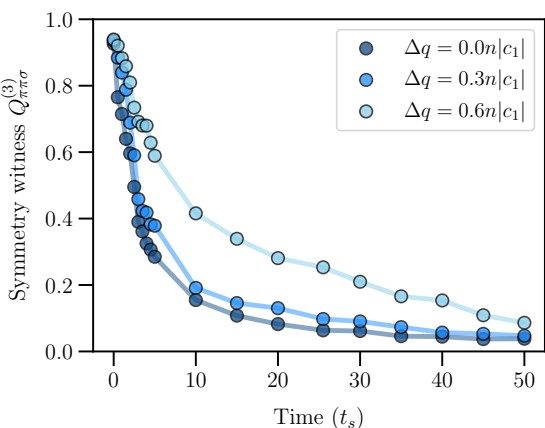

Figure 4: Evolution of the symmetry witness $Q^{(3)}_{\pi\pi\sigma}$ for three different systems prepared in a symmetry-broken state. The dark blue curve represents the symmetry witness for a system that is not quenched initially, while the mid- and lightblue curves correspond to initial quenches, with the lightblue one being a stronger quench. The middle curve for $Q^{(3)}$ shows the same data as in Fig. 3, but only up to $50\,t_s$.

where the correlation functions are typically larger, we normalize the difference by dividing it by double the average value of $|f^{(3)}|$ and $|f^{(2)}|$, which yields

$$Q^{(3)}_{\pi\pi\sigma}(t) = \lim_{\varepsilon \to 0^+} \left\Vert \frac{f^{(3)}_{t,\pi\pi\sigma} - f^{(2)}_{t,\pi\pi\sigma}}{\left| f^{(3)}_{t,\pi\pi\sigma} \right| + \left| f^{(2)}_{t,\pi\pi\sigma} \right| + \varepsilon} \right\Vert . \tag{21}$$

Here, $\varepsilon$ is a regularization parameter ensuring that $Q^{(3)}_{\pi\pi\sigma} = 0$ when $f^{(3)}_{t,\pi\pi\sigma} = f^{(2)}_{t,\pi\pi\sigma} = 0$, i.e., in the absence of both explicit as well as spontaneous symmetry breaking. In practice, the choice of $\varepsilon$ is motivated by the value of statistical error, inevitable in any experimental or numerical setup. Note that the normalization choice implies $0 \leq Q^{(3)}_{\pi\pi\sigma} \leq 1$, with the upper bound following from the Cauchy–Schwarz inequality.

At each point in time, the quantity $Q^{(3)}_{\pi\pi\sigma}$, which we call a symmetry witness, connects one-, two-, and three-point correlation functions and quantifies the degree of violation of the symmetry identity (18). Analogously, one can introduce higher-order witnesses $Q^{(4)}_{\pi\pi\sigma\sigma}$ and $Q^{(4)}_{\pi\pi\pi\pi}$ using the identities (19a) and (19b), respectively, characterizing the symmetry content with respect to the higher-order correlation functions. Geometrically, the connected correlation functions characterize the shape and the inner structure of the histograms like the ones depicted in Fig. 2. Such histograms consist of "sub-histograms", one for each spatial point $x_i$, or momentum mode $p_i$, in the system. The one-point functions correspond to their positions, the two-point functions are related to their widths and heights, while higher-order $n$-point functions reflect cross-correlations between the sub-histograms. Symmetry then puts constraints on their allowed shapes and cross-correlations, and symmetry witnesses represent how well these constraints are satisfied. The spatial correlation functions can be extracted from numerical simulations or experimentally by sampling read-outs of the transverse spin $F_\perp(x) = F_x(x) + iF_y(x)$ [22]. As a result, probing symmetry properties of the system via exact relations between observable correlation functions proves to be an effective approach, as demonstrated in the following sections.

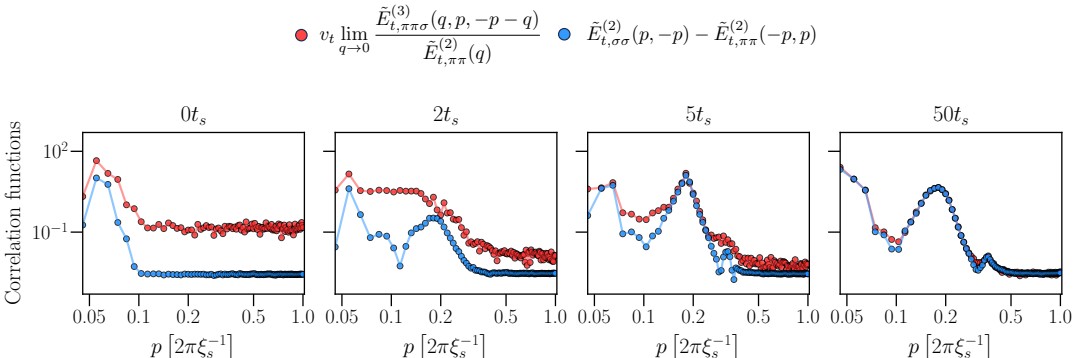

Figure 5: Data for the symmetry identity (18) with the correlation functions as a function of momentum at four different times during the dynamical evolution.

# 5 Nonequilibrium symmetry restoration

In the following, we investigate the dynamics of a spinor Bose gas (4) prepared in an explicitly symmetry-broken state. Whether the initially explicitly broken symmetry gets effectively restored during the dynamics will be analyzed using the symmetry witnesses introduced above. We employ the truncated Wigner approximation (TWA), which describes the dynamics for highly occupied systems at not too late times and weak couplings [25]. The numerical integration of the system is done via a pseudo-spectral split-step method and gives the time evolution of the full spinor state $\boldsymbol{\psi} = (\psi_1, \psi_0, \psi_{-1})^T$ comprised of the complex scalar Bose fields describing the three magnetic components of the spin-1 manifold.

We start from an initial state with nonvanishing $n$-point spin correlations that violate the SO(2) rotational symmetry in the $F_x - F_y$ plane. For this we consider the spinor condensate in the mean-field ground state of the easy-plane phase,

$$\boldsymbol{\psi}_{\text{EP}} = \frac{e^{i\theta}}{2} \begin{pmatrix} e^{-i\varphi_{\text{L}}/2}\sqrt{1-q/2\tilde{q}} \\ \sqrt{2+q/\tilde{q}} \\ e^{i\varphi_{\text{L}}/2}\sqrt{1-q/2\tilde{q}} \end{pmatrix}, \tag{22}$$

which is characterized by a well-defined spin length and orientation. In addition, we imprint a Gamma distribution function in momentum space in the fundamental fields and add noise in the Bogoliubov modes of the initial state to achieve a sizeable explicit symmetry breaking. We then quench the quadratic Zeeman shift from $q_{\text{i}} = 0.9n|c_1|$ to $q_{\text{f}} = 0.6n|c_1|$, where we verified that no significant excitations of topological defects are excited in the system. We propagate this state according to the classical field equations of motion,

$$i\partial_t \boldsymbol{\psi}(x,t) = \left[ -\frac{1}{2M}\frac{\partial^2}{\partial x^2} + qf_z^2 + c_0\,n(x,t) + c_1\boldsymbol{F}(x,t)\cdot\boldsymbol{f} \right]\boldsymbol{\psi}(x,t), \tag{23}$$

with periodic boundary conditions.

The physical parameters of the simulations aim to resemble a cloud of $^{87}$Rb atoms in a one-dimensional geometry as performed in the experiments [7, 23, 26], the main differences being an increased homogeneous density $n$ compared to the experiment and a purely one-dimensional setting with no trapping potential. We simulate a cloud of $3 \cdot 10^6$ particles on a numerical grid containing $N = 4096$ points corresponding to a physical length of $220\,\mu\text{m}$. The spin healing length is given by $\xi_{\text{s}} = 8$ lattice units, and spin-changing collisions occur on a timescale of $t_{\text{s}} = 696$ in numerical time units. We give spatial length in terms of the spin healing length $\xi_{\text{s}} = (2Mn|c_1|)^{-1/2}$ and time in units of the characteristic spin-changing

collision time $t_s = 2\pi/(n|c_1|)$. Furthermore, the field operators are normalized with respects to the total density $\tilde{\psi}_m = \psi_m/\sqrt{n}$, which results in a normalization of the spin vector as well $\tilde{F} = F/n$. In the following, the tilde is omitted and all values are to be understood as normalized values unless explicitly stated otherwise. Upon extracting the spin degrees of freedom $F_x$ and $F_y$, we compute the relevant two-, three-, and four-point correlation functions appearing in the identities (18), (19a), and (19b). Further technical details of these computations are given in App. C.

It is instructive to first examine the probability distribution of local spins in real space by averaging over many realizations. In Fig. 2, we depict an $F_z = 0$ cut of the probability density in spin configuration space. From the left graph, one observes that the initial state is characterized by a sizeable spin length with a rather well-defined orientation. As a consequence, one may separate two types of excitations for the transversal spin $F_\perp$: a radial "Higgs"-like mode associated with perturbations of the spin length $|F_\perp|$, and a transverse "Goldstone"-like mode associated with perturbations of the angle $\varphi_L$, respectively. Since the state is initialized away from the minimum of the sombrero-shaped effective potential illustrated in Fig. 1, one observes dynamics in the radial direction, such that the spin length $|F_\perp|$ acquires a range of values which are also significantly smaller than the initial one. As seen in the histograms, this occurs predominantly during the first few characteristic spin-changing collision times $t_s$. During this time, the nonequilibrium "Higgs"-like mode explores the inner part of the effective potential, whose nonconvex shape is expected to lead to a fast instability growth of the mode occupancy in a characteristic momentum range. However, after about $\sim 5\, t_s$, perturbations in $|F_\perp|$ are seen to become more and more suppressed. Instead, significantly slower dynamics for the transverse mode starts dominating, by which the spin distribution settles into a banana-like shape as it spreads out around the ring set by the minimum of the effective potential.

While the histograms indicate the different dominant excitations and timescales of the system, one needs further information to quantify the initial explicit symmetry breaking and its effective restoration. For instance, both the left graph of Fig. 2 at $0\, t_s$ and the right one at $100\, t_s$ indicate configurations with comparable spin length and rather small spread in the radial direction. However, their transverse extensions along the ring, which represent the "Goldstone"-like fluctuations, are significantly different. As described in Sec. 4, in the absence of explicit symmetry breaking there exists a well-defined relation between the spin length and the fluctuations, which we will use in the following to quantify the symmetry content of the data.

Fig. 3 shows the corresponding time evolution of the symmetry witnesses $Q^{(n)}$ defined in Eq. (21), where $0 \leq Q^{(n)} \leq 1$, with $Q^{(n)} = 0$ in the absence of explicit symmetry violation. The index $n$ denotes the maximum number of spatial points involved in the correlation functions probing the symmetries. We show $Q^{(3)}_{\pi\pi\sigma}$ based on an identity connecting two- and three-point functions involving the "Goldstone"-like ($\pi$) and "Higgs"-like ($\sigma$) excitations appearing in Eq. (18), while $Q^{(4)}_{\pi\pi\sigma\sigma}$ and $Q^{(4)}_{\pi\pi\pi\pi}$ connect three- and four-point functions based on Eqs. (19a) and (19b), respectively.

As seen in Fig. 3, the systems starts out in a state that explicitly breaks the SO(2) symmetry of the underlying Hamiltonian very strongly, with the different $Q^{(n)}$ rather close to unity. While the unitary time evolution of the quantum system can never restore the symmetry exactly, one observes that important observable properties can nevertheless exhibit an effective symmetry restoration. The different witnesses based on $n$-point correlation functions probe more and more details as $n$ increases. Correspondingly, we find that the lowest-order witness shown, $Q^{(3)}_{\pi\pi\sigma}$, approaches zero fastest (blue curve). In fact, after an initial rapid decrease until times of a few $t_s$, the restoration dynamics slows down, and the timescales are in close analogy to those observed from the histograms in Fig. 2.

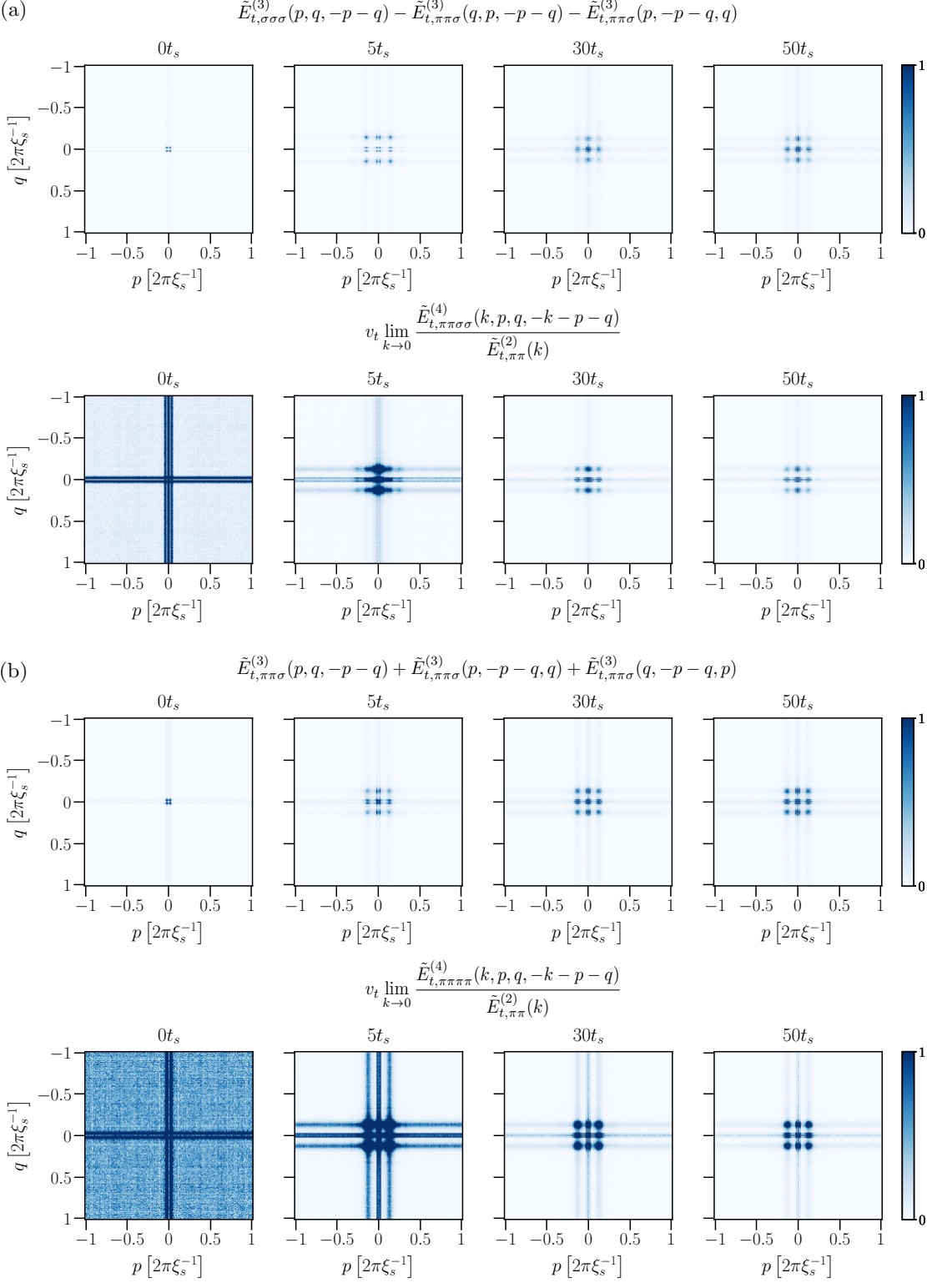

Figure 6: Momentum-conserving surfaces in the symmetry identities (19a) and (19b), respectively.

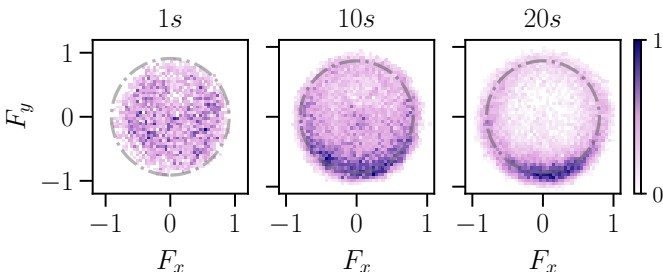

Figure 7: Histograms of the experimentally measured spin in the $F_x - F_y$ plane taken from a quasi-one-dimensional $^{87}$Rb experiment [23], normalized by the atom number, for different evolution times. The dash-dotted line represents $|F_\perp| = 0.85$.

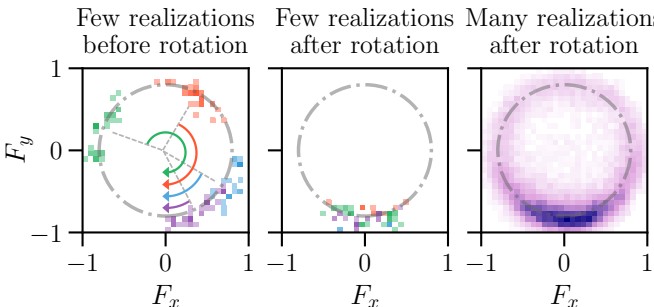

Figure 8: Histogram of the normalized spin at $t = 35$s. The left and middle figures show four different experimental realizations, each in different color, before (left) and after (middle) the rotation by the mean phases for each realization. On the right, the combination of all realizations with mean phase subtracted is displayed.

The higher-order witnesses $Q^{(4)}_{\pi\pi\sigma\sigma}$ (green curve) and especially $Q^{(4)}_{\pi\pi\pi\pi}$ (red curve) exhibit a comparably slower effective restoration of the initially broken symmetry. While $Q^{(4)}_{\pi\pi\sigma\sigma}$ involving both $\sigma$ and $\pi$ excitations still shows a characteristic two-stage decay, which is relatively fast at early times and then slowing down at late times, this is much less pronounced in $Q^{(4)}_{\pi\pi\pi\pi}$, which involves predominantly the slow "Goldstone"-like modes. Nevertheless, all witnesses clearly exhibit the approach towards an effective restoration of the explicitly broken symmetry by the initial state. We emphasize that this is much shorter than the timescale on which the approach to thermal equilibrium is observed, as the power spectrum $\langle |F_\perp|^2 \rangle$ starts to develop a thermal tail at higher momenta around $\sim 1400\, t_s$. This separation of time scales between the effective restoration of an explicitly broken symmetry and thermalization may, in principle, be further diminished for sufficiently high-order correlation functions. However, thermalization time is defined with respect to characteristic thermodynamic observables that typically do not involve arbitrarily high-order details since the time-translation invariant thermal state can never be reached on a fundamental level in systems with unitary dynamics. In practice, emergent theories that effectively describe dynamical behavior, such as effective kinetic theories, are based on a reduced set of low-order correlation functions. In this context, our results demonstrate that effective symmetry restoration can occur long before the system equilibrates. The situation is reminiscent of thermalization in isolated quantum systems, where local observables of the system, prepared in a nonequilibrium quantum state, eventually behave as if sampled from a thermal distribution. Similarly, while low-order symmetry witnesses show effective restoration, some higher-order witnesses, which encode finer statistical details of the system, will show symmetry violations at asymptotically late times. This is in accordance

with the general statement regarding how the symmetry can never be fully restored by means of a unitary time evolution governed by a symmetric Hamiltonian, cf. Sec. 2.

It remains to investigate to what extent the results depend on the details of the initial state. Here we consider variations in the initial quench of the quadratic Zeeman shift with different strengths, or with no quench at all. As depicted in Fig. 4, we find that the stronger the quench, the longer it takes to restore the SO(2) symmetry, and not quenching at all restores it the fastest. The witness based on the correlation functions from Eq. (18), as seen in Fig. 3, corresponds to the middle curve, with an initial quench from $q_i = 0.9n|c_1|$ to $q_f = 0.6n|c_1|$. Quenching stronger than this, to $q_f = 0.3n|c_1|$, takes longer to restore the symmetry (light blue curve), and not doing a quench takes the shortest (dark blue curve). Irrespective of the strength of or the presence of the quench, the correlation functions and the restoration process look qualitatively very similar as shown in Fig. 4.

The symmetry witnesses provide an efficient means to quantify the symmetry content of the data. However, further details can be investigated by looking directly at the underlying momentum-resolved correlation functions in the identity (18). In Fig. 5, we plot both the left-hand side (red curve) and right-hand side (blue curve) of Eq. (18) for four different time steps. Initially, we observe that the symmetry is strongly broken signalled by the unequal different $n$-point correlation functions. Within the span of a few $t_s$, these different correlation functions quickly approach each other and by $\sim 50\,t_s$, they are nearly equal and the conclusions are as for the symmetry witnesses discussed before. In addition, one observes from the momentum-resolved correlation functions that, apart from the initial strong fluctuations at low momenta, an additional peak in the correlation functions develops at a higher momentum scale. The peak height settles quickly within a few $t_s$, during which the "Higgs"-like mode explores the inner part of the effective potential leading to a fast growth of fluctuations as discussed above.

The momenta of the correlation functions entering the identity (18) underlying $Q^{(3)}$ correspond to the momentum-conserving diagonals of the full momentum matrix. Likewise, the identities for $Q^{(4)}$ involve momentum-conserving surfaces. As an example, we show the surfaces of our numerical data corresponding to the symmetry identities (19a) in Fig. 6(a) and (19b) in Fig. 6(b). In both cases, we see strong initial symmetry violation signalled by the different unequal $n$-point correlator surfaces. The cross-like shape is the dominant feature of these surfaces and is already present initially, although much stronger in the four-point surfaces. The appearance of the surfaces becomes gradually more equal with time in both Fig. 6(a) and (b), however, we can visually confirm that it is not as quick as for the momentum-conserving diagonals above. Additionally, restoration is visibly slower for the identity (19b) since at $50\,t_s$ in Fig. 6(b) the dominant cross-like features are still at an increased amplitude in the four-point surface compared to the three-point one. This is consistent with what we have observed from the corresponding witnesses in Fig. 3.

# 6 Nonequilibrium spontaneous symmetry breaking

In the previous section, we discussed the explicit breaking of a symmetry of the Hamiltonian by the initial state, and its effective restoration long before the system equilibrates. However, even if explicit symmetry breaking is absent or dynamically restored, the symmetry may still be spontaneously broken. The notion of spontaneous symmetry breaking, in thermal equilibrium or dynamically even far from equilibrium, is a central ingredient for our understanding of phase transitions as explained in Sec. 2. Spontaneous symmetry breaking is signalled by a nonzero order parameter (2) using a bias that does not break the symmetry explicitly in the end.

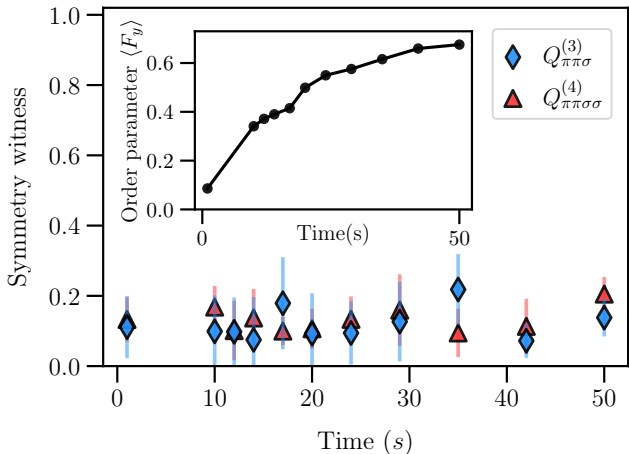

Figure 9: The symmetry witness based on two- and three-point correlation functions extracted from experimental data. The inset shows the average spin length $\langle F_y \rangle$. The witness $Q^{(4)}_{\pi\pi\pi\pi}$, which is not shown in order not to overcrowd the plot, gives comparable results to $Q^{(4)}_{\pi\pi\sigma\sigma}$. The error bars are obtained by bootstrapping and correspond to 80% confidence interval. The spin-changing collision time is $t_s = 2\pi/(n|c_1|) \sim 0.4\,\mathrm{s}$ for the experimental parameters used in this work.

To analyze spontaneous symmetry breaking out of equilibrium in more detail, in the following we consider experimental data from measurements of a spinor Bose–Einstein condensate of $^{87}$Rb atoms as described in Sec. 3. The system is initialized in the $|F, m_\mathrm{F}\rangle = |1, 0\rangle$ state, the so-called polar state. Subsequently, the parameter $q$, which corresponds to the relevant energy difference between the $m_\mathrm{F} = 0$ and $m_\mathrm{F} = \pm 1$ levels, is quenched to a value within the easy-plane phase thereby initiating the dynamics. In contrast to the initial state investigated in Sec. 5 in the context of explicit symmetry breaking, in the present case there is initially no well-defined spin length, with fluctuations solely in the $F_x - F_y$ plane such that the initial state respects the SO(2) symmetry of the system. The initial conditions restrict the average longitudinal ($z$-axis) spin to be zero, and excitations build up in the $F_x - F_y$ plane. This transversal spin degree of freedom is examined by the spatially resolved detection of the complex-valued field $F_\perp(x) = F_x(x) + \mathrm{i}F_y(x)$ [23].

Fig. 7 shows histograms of the measured spin orientations in the $F_x - F_y$ plane normalized by the atom number, at different times. While initially the measured values scatter, such that the average spin length is practically zero, this changes at later times. The average spin length settles around $|F_\perp| = 0.85$ represented by the dash-dotted line in the figure. In this case, the nonzero average spin plays the role of the order parameter signalling the spontaneous symmetry breaking of the SO(2)-symmetric system. Due to the underlying SO(2) symmetry, one can always align the expectation value along one of the axes, e.g., $\langle \hat{F}_x \rangle = 0$, $\langle \hat{F}_y \rangle = v_t$, which was done for Fig. 7.

The alignment procedure of the spin expectation value for the experimental data is illustrated in Fig. 8. In the left graph, data from four experimental realizations is shown at late time ($t = 35$s). To understand the underlying dynamics leading to these configurations, it is helpful to consider them as corresponding to the top view of the pictorial representation of the sombrero effective potential sketched in Fig. 1(c). While in each realization the spin distribution is expected to acquire a "blob" shape, as marked by red in the green ring of that figure, and settle in one of the many symmetry-breaking minima, many such blobs will form a symmetric ring. Hence, while there is a preferred direction in each experimental realization individually, once we average over multiple realizations, the transverse spin is symmetrically distributed

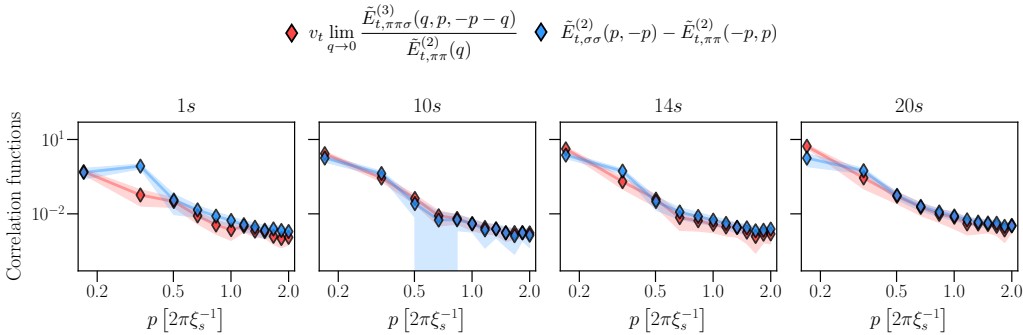

Figure 10: The left- and right-hand sides of the symmetry identity (18) using experimental data, with the momentum-resolved correlation functions at four different times during the dynamical evolution. The error bands represent 80% confidence intervals obtained from bootstrapping.

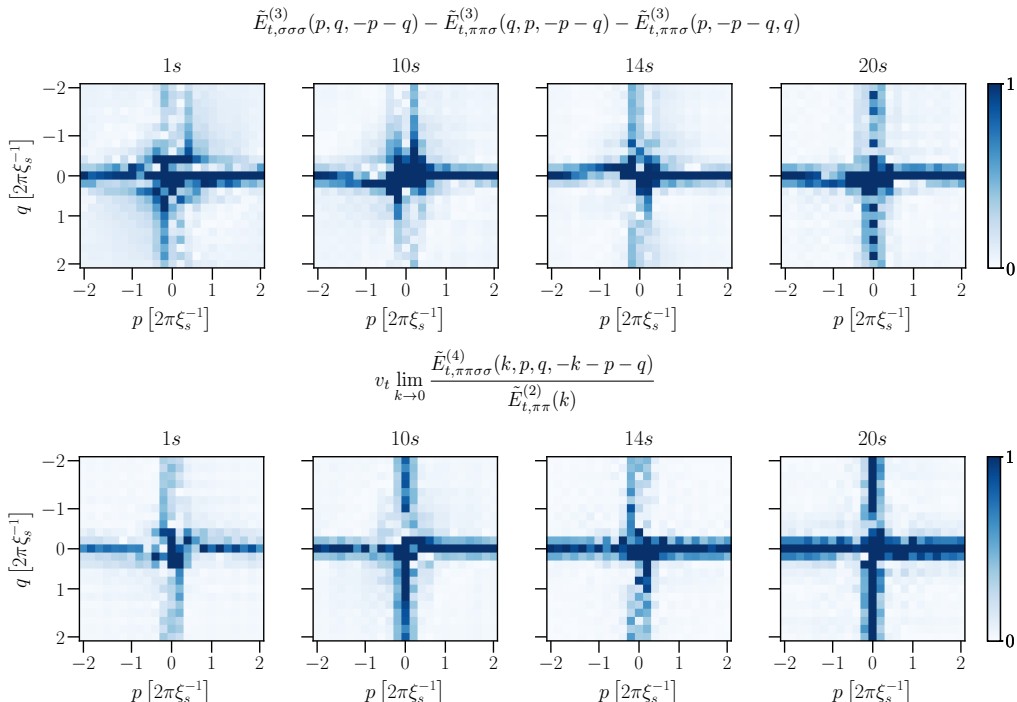

Figure 11: Data for the symmetry identity connecting two-, three-, and four-point correlation functions calculated from experimental measurements. The top four surface plots correspond to the right-hand side of the identity (19a), while the bottom ones correspond to the left-hand side of the equation. One observes the resemblance of these momentum-conserving surfaces, which involve different $n$-point correlation functions.

across the ring in the $F_x - F_y$ plane. Correspondingly, one observes the different experimental realizations distributed along the ring as seen in the left graph of Fig. 8. However, by rotating each individual realization by the global phase as shown in the middle of Fig. 8, there is a nonzero expectation value $\langle \hat{F}_y \rangle \neq 0$ and $\langle \hat{F}_x \rangle = 0$ when averaged over all the realizations. This is shown in the right graph of Fig. 8, which gives the average over many realizations. We emphasize that the global-phase rotation angle maintains translational invariance since this angle does not introduce any spatial bias, whereas, e.g., rotating by the phase of any specific point would do so.

While the histograms of Fig. 7 and 8 illustrate the dynamical build-up of a macroscopic spin length, a quantitative analysis of spontaneous symmetry breaking requires taking its fluctuations into account as well. In particular, the fluctuations can be used to distinguish data with underlying spontaneous symmetry breaking from situations where a macroscopic spin length arises due to explicit symmetry breaking, as exemplified on the left of Fig. 2. The fluctuations are encoded in the $n$-point correlation functions, which fulfill the symmetry identities for spontaneous symmetry breaking as derived in Sec. 4.

We examine the witnesses $Q^{(3)}_{\pi\pi\sigma}$ and $Q^{(4)}_{\pi\pi\sigma\sigma}$ according to Eq. (21) in Fig. 9. The minimum value of these quantities, and any of the higher-order witnesses is zero, which corresponds to a perfectly symmetric scenario including that of a spontaneously broken symmetric state, while the upper value is unity corresponding to a maximally and explicitly broken state. One observes that the value of the symmetry witnesses is clearly much smaller than unity, and near zero within errors. This indicates the absence of explicit symmetry breaking, which in principle can be improved with increasing statistics. We also give the average spin length $\langle F_y \rangle$ as an inset on top of the symmetry witness. The witness is seen to be near zero within errors independent of the magnitude of $\langle F_y \rangle$. One observes that the magnitude of $\langle F_y \rangle$ settles at later times, representing an order parameter for spontaneous symmetry breaking.

In order to test the momentum resolved symmetry identity (18), we consider the two- and three-point correlation functions by averaging over many realizations of single-shot measurements of the rotated $F_\perp(x)$. For more details on the data analysis procedure, see App. C. We plot four different time steps in Fig. 10 and observe that the left- and the right-hand sides of the identities are close within experimental errors at all times. Similarly, Fig. 11 shows momentum resolved surface plots for the symmetry identity (19a) connecting two-, three-, and four-point correlation functions calculated from experimental measurements. We emphasize once again that *a priori* there is no reason why these different $n$-point correlation functions should obey such equalities, representing a quantitative manifestation of the emergence of spontaneous symmetry breaking.

# 7 Discussion and outlook

While symmetries of a Hamiltonian that are explicitly broken by the initial state cannot be restored on a fundamental level in closed quantum systems, we have shown that their effective restoration can be quantified in terms of symmetry identities for correlation functions. In particular, our results demonstrate that properties involving lower $n$-point correlation functions exhibit dynamical symmetry restoration earlier than those involving higher-order correlations. Moreover, our findings for a spinor Bose gas show that an initial explicit symmetry breaking gets restored on timescales much before the system thermalizes. These are important ingredients for effective descriptions of nonequilibrium evolutions, which are typically based on lower-order correlation functions, where kinetic theory or Boltzmann equations for single-particle distribution functions extracted from two-point correlation functions represent a paradigmatic example [27].

Though the correlation functions appearing in the symmetry identities (18), (19a), and (19b) involve only few spatial points, in general they also test extremely nonlocal properties, such as the ones encoded in their low-momentum behavior in Fourier space. This is crucial for the identification of spontaneous symmetry breaking in the presence of a nonvanishing expectation value for the zero mode and condensation phenomena, which we have analyzed for the example of the spinor Bose gas. In particular, our approach is not based on a spatial separation into subsystems, which can be difficult to define in fundamental descriptions, such as relativistic theories, and gauge theories implementing local symmetries. Though we have

not described the approach for local symmetries explicitly in this work, the formulation of nonequilibrium (equal-time) versions of Ward identities for gauge theories [12,28–30] follows along the same lines as we described.

Our approach provides a general pathway to extract the symmetry content of nonequilibrium quantum as well as classical many-body systems based on a hierarchy of $n$-point correlation functions. This complements alternative approaches to the question of dynamical symmetry restoration, such as the entanglement asymmetry between spatial subsystems introduced as a measure of symmetry breaking in quantum systems [9, 10, 31–37], which has also been experimentally applied [38–40]. It would be interesting to establish a direct link between our symmetry witnesses based on correlations and the entanglement measure of symmetry breaking for quantum systems. While our work primarily focused on ultracold atoms, the approach could also give important further insights into applications and experimental data across various systems, ranging from the detection of new nonequilibrium phases in condensed matter systems to preheating dynamics in inflationary early-universe cosmology [1–3].

## Acknowledgments

The authors thank Y. Deller, T. Gasenzer, P. Heinen, S. Lannig, M. Prüfer, D. Spitz, and T. V. Zache for discussions on the topics described here.

**Funding information**  This work is part of and funded by the Deutsche Forschungsgemeinschaft (DFG, German Research Foundation) under Germany's Excellence Strategy EXC 2181/1-390900948 (the Heidelberg STRUCTURES Excellence Cluster) and the Collaborative Research Centre, Project-ID No. 273811115, SFB 1225 ISOQUANT. The authors acknowledge support by the state of Baden-Württemberg through bwHPC and DFG through grant INST 35/1597-1 FUGG. A.N.M. acknowledges support by QuantERA II Programme that has received funding from the European Union's Horizon 2020 Research and Innovation Programme under Grant Agreement No 101017733 ("QuSiED") and by the Dynamics and Topology Center funded by the State of Rhineland Palatinate.

## A   Experimental details and analysis

For our analysis, we use data obtained with a $^{87}$Rb spinor BEC of $\sim 10^5$ atoms in the $F = 1$ hyperfine manifold with initial state $|F, m_F\rangle = |1, 0\rangle$. The atom cloud is contained in a quasi one-dimensional trapping geometry, which consists of a dipole trap formed by a 1030 nm laser beam with trapping frequencies $(\omega_\parallel, \omega_\perp) = 2\pi \times (1.6, 160)$ Hz, and with two end caps formed by beams at 760 nm, confining the atoms within the central part of the harmonic potential. The longitudinal harmonic potential is constant to a good approximation over the employed sizes, leading to a 1D box-like confinement, with size $\sim 100$ μm in the measurements used. The atom cloud is subjected to a uniform magnetic field of $B = 0.894$ G throughout the experiment which leads to a quadratic Zeeman splitting of $q_B \sim h \times 58$ Hz. The spin dynamics is controlled via off-resonant microwave dressing $q = q_B + q_{MW}$ with $q < 2n|c_1|$. The initial quench is implemented by the instantaneous switching on of the microwave power.

The transverse spin field $F_\perp = F_x + iF_y$ readout is obtained via spin rotations and microwave coupling to the initially empty $F = 2$ hyperfine manifold prior to a Stern–Gerlach pulse and spatially resolved absorption imaging. For a more detailed account on the experimental setup and on how the measurements were obtained, see the supplementary material of Ref. [23]. While the spatial degree of freedom is continuous, it gets discretized in the analysis procedure

Table 1: Number of experimental realizations.

| Evolution time (s) | Number of realizations |
|:---:|:---:|
| 1 | 68 |
| 10 | 237 |
| 12 | 236 |
| 14 | 237 |
| 17 | 236 |
| 20 | 239 |
| 24 | 238 |
| 29 | 269 |
| 35 | 296 |
| 42 | 298 |
| 50 | 296 |

by the finite pixel size of the camera and imaging resolution ($\approx 1.2\,\mu$m per three pixels). Our analysis focuses on the central $\sim 100$ pixels of the data since, as discussed in the main text in Sec. 3, establishing long-range coherence across the entire system requires some time.

## B  Physical interpretation of the symmetry breaking perturbation

Since the spin operators $\hat{F}_i$ are the generators of the rotational symmetry, they commute with a symmetric Hamiltonian and consequently with the evolution operator as well. This allows us to rewrite the generating functional as

$$
\begin{aligned}
Z_t[\boldsymbol{J}] &= \mathrm{Tr}\Big\{\mathcal{U}(t,t_0)\,\mathrm{e}^{\int \mathrm{d}x\boldsymbol{J}(x)\cdot\hat{\boldsymbol{F}}(x)/2}\,\hat{\rho}_{t_0}\,\mathrm{e}^{\int \mathrm{d}x\boldsymbol{J}(x)\cdot\hat{\boldsymbol{F}}(x)/2}\,\mathcal{U}^\dagger(t,t_0)\Big\} \\
&= \mathrm{Tr}\Big\{\mathcal{U}(t,t_0)\,\hat{\rho}'_{t_0}(\boldsymbol{J})\,\mathcal{U}^\dagger(t,t_0)\Big\},
\end{aligned}
\tag{B.1}
$$

where we have introduced the deformed initial density matrix

$$
\hat{\rho}'_{t_0}(\boldsymbol{J}) \equiv \mathrm{e}^{\int \mathrm{d}x\boldsymbol{J}(x)\cdot\hat{\boldsymbol{F}}(x)/2}\,\hat{\rho}_{t_0}\,\mathrm{e}^{\int \mathrm{d}x\boldsymbol{J}(x)\cdot\hat{\boldsymbol{F}}(x)/2}.
\tag{B.2}
$$

Note that, provided the sources $J_i$ are real, the deformed operator $\hat{\rho}'_{t_0}(\boldsymbol{J})$ is Hermitian. Furthermore, under the same condition, it is also positive semidefinite. Indeed,

$$
\langle\psi|\,\hat{\rho}'_{t_0}(\boldsymbol{J})\,|\psi\rangle = \langle\psi_{\boldsymbol{J}}|\,\hat{\rho}_{t_0}\,|\psi_{\boldsymbol{J}}\rangle \geq 0,
\tag{B.3}
$$

with $|\psi_{\boldsymbol{J}}\rangle \equiv \mathrm{e}^{\int \mathrm{d}x\boldsymbol{J}(x)\cdot\hat{\boldsymbol{F}}(x)/2}|\psi\rangle$, and $\hat{\rho}_{t_0}$ is positive semidefinite being a density matrix by assumption. Thus, aside from normalization, $\hat{\rho}'_t$ satisfies all the conditions of a physical density matrix. This suggests a simple interpretation of the equal-time generating functional $Z_t[\boldsymbol{J}]$ in the absence of explicit symmetry violations: it represents the evolution of the symmetric density matrix $\hat{\rho}_{t_0}$ that has been deformed by means of linear sources coupled to the spin operators $\hat{F}_i$ at the initial time $t_0$, thus breaking the symmetry.

Let us remark that the above simple physical picture is, to a certain extent, unique for spin systems. The reason is that the linear-source term that enters the definition of the generating functional $Z_t[\boldsymbol{J}]$ and serves as a symmetry-breaking perturbation commutes, in this case, with the symmetric evolution operator $\mathcal{U}$ as the spin operators $\hat{F}_i$ are also generators of the symmetry group.

Nevertheless, provided the linear source $\boldsymbol{J}$ in the definition of $Z_t[\boldsymbol{J}]$ is coupled to operators that transform nontrivially under the symmetry group in question, the formalism developed in this work can still be applied to define spontaneous symmetry breaking in nonequilibrium systems, albeit lacking the appealing interpretation of the symmetry-breaking perturbation as a deformation of the initial state.

## C  Calculation of correlation functions

Both experimentally and in TWA simulations, we have $N_s$ samples (measurements) of the spin observable $F_i$ in datasets $\left\{F_i^{(s)} \mid s = 1, \dots, N_s\right\}$, from which we infer $n$-th order correlation functions as

$$\langle F_{i_1} \cdots F_{i_n} \rangle \approx \frac{1}{N_s} \sum_{s=1}^{N_s} F_{i_1}^{(s)} \cdots F_{i_n}^{(s)}. \tag{C.1}$$

The information in all of the $n$-point correlation functions is equivalently stored in the generating functional $Z[J]$ as described in the context of Eq. (6). The TWA simulations involve periodic boundary conditions, and while the experimental setup considered is a finite system without periodic boundary conditions, we find approximate translational invariance, which simplifies the calculation of connected correlators in momentum space. We first perform a discrete Fourier transform (DFT) for the spin observables $F_i$ to momentum space

$$F_i^{(s)}(p) = \mathrm{DFT}_{x \to p}\left[F_i^{(s)}(x)\right] \equiv \sum_{j=1}^{N} e^{-ipj} F_i^{(s)}(j), \tag{C.2}$$

where $p \in [p_L, 2p_L, \dots, Np_L]$, $p_L = 2\pi/L$, and $L$ is the system size. Subsequently, we compute connected correlation functions in momentum space using the Julia language package Cumulants.jl [41].

We have verified that this procedure gives equivalent results to first computing connected correlators in position space, and then performing the DFT. The former approach, however, is much more memory-efficient. Indeed, computing higher-order correlation functions requires a considerable amount of computer memory: for instance, a four-point cumulant is an $N \times N \times N \times N$ array, so the amount of required memory scales quartically with the system size. At the same time, as evident from Eqs. (19a) and (19b), the four-point functions entering the symmetry identities have one of the momenta set to zero while the three remaining ones have to add up to zero due to momentum conservation. Therefore, one only needs a two-dimensional momentum-conserving surface, which can be encoded in an $N \times N$ matrix. By computing correlators directly in momentum space we avoid the need to store the full $N \times N \times N \times N$ array, and we can directly extract the relevant information by computing the two-dimensional momentum-conserving surface. For our numerical data, we consider correlation functions up to the inverse healing length, where the TWA description is expected to be reliable. For the plots, we have binned every 5 data points, while the correlators themselves were calculated on uncoarsened lattices.

Note that since perfect homogeneity and isotropy cannot be experimentally achieved, numerical artefacts always enter analyses. More specifically, in Eq. (18), while $E_{\pi\pi}^{(2)}(-p, p)$ and $E_{\sigma\sigma}^{(2)}(p, -p)$ are manifestly real, the three-point function $E_{\pi\pi\sigma}^{(3)}(0, p, -p)$ has in general a nonzero imaginary part. However, for the experimental data, the imaginary part is orders of magnitude below the real part, therefore the magnitude of the correlator is dominated by the contribution from the real part. We similarly observe this with numerical data, apart from the very early initial times of a few $t_s$, where the imaginary part is more pronounced. In this case, and in all other cases, the magnitude of complex quantities is plotted.

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
