# Peer review of "Extracting the symmetries of nonequilibrium quantum many-body systems"

_SciPost Physics, doi:SciPost Phys. 18, 044 (2025)_

## Round 1 · Referee Report · Anonymous (Referee 1) · 2024-11-12

Strengths

1. Novelty and Relevance: The work addresses symmetry properties in nonequilibrium quantum systems, especially where symmetry restoration dynamics play a critical role. The methodology, focusing on correlation functions as symmetry indicators, is innovative and well-motivated.

2. Analytical Rigor and Theoretical Foundations: The derivation of symmetry identities and the development of symmetry witnesses are grounded in a clear, structured theoretical framework. The paper also provides detailed derivations and theoretical justifications that enhance its scientific rigor.

3. Broad Applicability: The framework is adaptable to a range of quantum many-body systems and is especially well-suited to experimental setups involving ultracold gases, as demonstrated by its application to spinor Bose gas data.

4. Use of Experimental and Numerical Validation: The authors support their approach with data from experimental setups and numerical simulations, which bolsters confidence in the method’s applicability and validity.

Weaknesses

1. Complexity of Presentation: Some sections, particularly those involving the derivation of symmetry identities (e.g., Eqs. (12)–(20)), could benefit from clearer, step-by-step explanations or additional diagrams to support reader comprehension. The technical complexity may obscure understanding for readers less familiar with the intricacies of Ward identities and symmetry witness construction.

2. Limited Experimental Comparison: While experimental data is provided, a more extensive discussion comparing theoretical predictions with experimental observations would improve understanding of the practical limits and performance of the proposed symmetry witness framework.

Report

The authors present an approach for identifying effective symmetries in nonequilibrium quantum many-body systems. This framework exploits equal-time correlation functions to distinguish symmetry properties of states from those of the Hamiltonian. The approach is applied to a spinor Bose gas, both through simulations and experimental data, to explore dynamic symmetry restoration and spontaneous symmetry breaking. The authors introduce "symmetry witnesses" based on correlation functions, a promising tool for characterising symmetry behaviour in quantum systems.

The manuscript presents an original and significant contribution to the study of symmetries in nonequilibrium quantum systems, offering an insightful methodology that could be impactful across quantum simulation and experiment domains. Given the strengths in innovation, theoretical robustness, and relevance to experimental physics, I recommend the paper for publication. Minor revisions, especially to enhance clarity in technical sections and to further clarify experimental implications, would be welcome.

Requested changes

  1. Clarification on Symmetry Witnesses: The definition and role of symmetry witnesses could be elaborated further to help readers grasp their significance, especially as they relate to experimentally measurable quantities. For instance, including a brief discussion on the intuition behind the choice of witnesses and how they correlate with physical observables could improve accessibility.

  2. The sentence: “The search for emergent theories that effectively de- scribe nonequilibrium macroscopic behavior, their classification and justification from first principles is one of the most pressing research directions in quantum many-body physics.” clearly needs to be supported by references. In general, I think the introduction can be supported by citations of relevant papers.

  3. Experimental Relevance: Expanding on how the experimental symmetry witness values were obtained (e.g., specific measurement methods or parameter choices in the Bose gas setup) would add clarity, as would a discussion on experimental noise or limitations that might affect symmetry witness accuracy.

  4. Further Examples of Applications: Discussing other potential applications, such as in cosmological simulations or condensed matter physics (which are mentioned by the authors), would highlight the broader utility of this methodology.

Recommendation

Publish (easily meets expectations and criteria for this Journal; among top 50%)

  • validity: top
  • significance: top
  • originality: top
  • clarity: top
  • formatting: perfect
  • grammar: perfect

Author:  Aleksandr Mikheev  on 2024-12-03  [id 5019]

(in reply to Report 1 on 2024-11-12)
Category:
answer to question

We would like to thank the referee for reviewing our paper and for the helpful comments.

The referee remarks: "Some sections, particularly those involving the derivation of symmetry identities (e.g., Eqs.(12)-(20)), could benefit from clearer, step-by-step explanations or additional diagrams to support reader comprehension."

Following the referee’s recommendation, we have added more steps in the derivation between Eqs. (12)-(20) in Sec. IV.

In addition, we have addressed the requested changes:

(1) We have added a brief discussion at the end of Sec. IV about the intuition behind the choice of witnesses and how they are connected to physical observables;
(2) Added some references in the introduction to support this;
(3) We have expanded on the specific measurement methods in Sec. III regarding the acquisition of experimental data and added more detailed parameter choices in App. A for the Bose gas setup. Since the error is dominated by finite sample size, a bootstrapping algorithm was used to obtain error estimates, as highlighted in the captions of Figs. 9-11;
(4) Other potential applications are now more specifically mentioned at the end of Sec. VII.

Anonymous on 2024-12-14  [id 5040]

(in reply to Aleksandr Mikheev on 2024-12-03 [id 5019])
Category:
answer to question

The authors have taken into account all my comments and made corresponding changes in the main text.
I recommend the publication in SciPost Physics.

---

## Round 1 · Referee Report · Anonymous (Referee 2) · 2024-11-15

Strengths

1-Adresses a relevant question that so far has received little attention
2-Introduces new tools that are of theoretical interest for the characterization of symmetries and can be implemented in current atomic experiments

Weaknesses

1-Sometimes ambiguous interpretation of symmetry witnesses (see report below for details)

Report

The manuscript introduces criteria that probe whether a given quantum state respects a specific type of symmetry. A framework is developed that can be applied to test the SO(2) symmetry of the spin dof of a spinor Bose-Einstein. From a moment generating function, correlation functions of different order are derived. This framework is applied to an SO(2) symmetric state whose symmetry-breaking can be revealed by applying a small bias field that afterwards tends to zero. The extent to which symmetry is broken is quantified in terms of so-called symmetry witnesses that are based on the L1 norm of the deviation from the fully symmetric expression. The methods are applied to theoretical and experimental data of spinor BECs.

Given the paramount importance of symmetries is all areas of modern quantum physics, the manuscript addresses an interesting and important question. The text is well written and the presentation is for the most part quite clear despite the rather technical content. Overall, I believe the work opens up an interesting avenue for further research and should be published. However, I have one concern that I believe should be addressed first:

My main concern is about the interpretation of the symmetry witnesses: If one such witness yields zero, one cannot conclude that the symmetry of the system is preserved/restored. It may well be that simply the low-order correlation functions do not reveal the symmetry-breaking but higher-order terms would show them. This is indeed reflected in the data discussed in the paper. What the analysis shows is simply that the observables are too simple to reveal the symmetry breaking.

On the one hand it is interesting that low-order correlations may appear to respect a symmetry that the full state does not respect. On the other hand, I believe that the statements about this aspect should be made more carefully: It is not the case that the symmetry has been restored on a shorter time scale than the approach to equilibrium, it is just not captured by this specific observable.

This is in direct analogy to entanglement witness from which the term is borrowed: Just because an entanglement witness (sufficient but not necessary condition for entanglement) yields zero, it does not imply that the state is separable. The interpretation is analogue to the case of Gaussian entanglement witnesses that restrict to first and second moments but fail to reveal the entanglement of strongly non-Gaussian states, which, in turn, can be captured by more sophisticated higher-order witnesses.

Requested changes

1-Revise the discussion of the interpretation of symmetry witnesses according to the comments above.

Recommendation

Ask for minor revision

  • validity: high
  • significance: high
  • originality: high
  • clarity: high
  • formatting: excellent
  • grammar: perfect

Author:  Aleksandr Mikheev  on 2024-12-03  [id 5020]

(in reply to Report 2 on 2024-11-15)
Category:
answer to question

We would like to thank the referee for the helpful comments.

**The referee remarks: **

If one such witness yields zero, one cannot conclude that the symmetry of the system is preserved/restored. It may well be that simply the low-order correlation functions do not reveal the symmetry-breaking but higher-order terms would show them.

Our response: We fully agree with this remark. The statement that the symmetry can never be fully restored is mentioned, e.g., in Sec. V of the manuscript:

"While the unitary time evolution of the quantum system can never restore the symmetry exactly, one observes that important observable properties can nevertheless exhibit an effective symmetry restoration. The different witnesses based on $n$-point correlation functions probe more and more details as $n$ increases."

**The referee says: **

It is not the case that the symmetry has been restored on a shorter time scale than the approach to equilibrium, it is just not captured by this specific observable.

Our response: Indeed, as implied in Sec. V of the manuscript,

"In practice, emergent theories that effectively describe dynamical behavior, such as effective kinetic theories, are based on a reduced set of low-order correlation functions. In this context, our results demonstrate that effective symmetry restoration can occur long before the system equilibrates."

the notion of symmetry, in this case, is observable-dependent. Effective symmetry restoration thus implies that the observables of interest, which typically consist of lower-order correlation functions, show no symmetry violation after a certain amount of time. The nonthermal character of the observed symmetry restoration mechanism is especially evident when analyzing the spectra, which start showing thermal features only at much later times, as mentioned in Sec. V of the manuscript:

"We emphasize that this is much shorter than the timescale on which the approach to thermal equilibrium is observed, as the power spectrum $\langle|F_{\perp}|^2\rangle$ starts to develop a thermal tail at higher momenta around $\sim 1400\,t_s$."

We thank the referee for pointing this out and in order to make things clearer, we have added a new paragraph in Sec. V:

"The situation is reminiscent of thermalization in isolated quantum systems, where local observables of the system, prepared in a nonequilibrium quantum state, eventually behave as if sampled from a thermal distribution. Similarly, while low-order symmetry witnesses show effective restoration, some higher-order witnesses, which encode finer statistical details of the system, will show symmetry violations at asymptotically late times. This is in accordance with the general statement regarding how the symmetry can never be fully restored by means of a unitary time evolution governed by a symmetric Hamiltonian, cf. Sec. II."

---

## Round 2 · Referee Report · Anonymous (Referee 2) · 2024-12-12

Report

The authors have addressed all concerns raised in my earlier report. With the added remarks I believe that there is no ambiguity about the interpretation of the symmetry witnesses. I recommend publication of the manuscript in its present form.

Recommendation

Publish (easily meets expectations and criteria for this Journal; among top 50%)

---

## Round 2 · Referee Report · Anonymous (Referee 1) · 2024-12-18

Report

The authors have taken into account all my comments and made
corresponding changes in the main text.
I recommend the publication in SciPost Physics.

Recommendation

Publish (easily meets expectations and criteria for this Journal; among top 50%)

---

## Round 2 · Author Response

Dear Editor,

We would like to thank the referees for their insightful reports. We provide a detailed response to their feedback in the replies below the reports. We hope that the revised version of our manuscript is suitable for publication in SciPost Physics.

---

## Round 2 · List of Changes

* Added references in the introduction.
* Added more clarity and details in Sec. III and App. A regarding the experiment and how measurements were obtained.
* Added more steps in the derivation between Eqs. (12)-(20) in Sec. IV.
* Added a discussion at the end of Sec. IV regarding the experimental connection.
* Revised the discussion of the interpretation of symmetry witnesses according to the comments of Referee #2 in Sec. V.
* Expanded on the discussion of potential applications at the end of Sec. VII.

---

## Editorial Decision

published